# Social Depolarization and Diversity of Opinions—Unified ABM Framework

**DOI:** 10.3390/e25040568

**Published:** 2023-03-26

**Authors:** Paweł Sobkowicz

**Affiliations:** Nomaten Centre of Excellence, National Centre for Nuclear Research, A Soltana 7, 05-400 Otwock, Poland; pawelsobko@gmail.com

**Keywords:** polarization, diversity, agent based models, opinion dynamics, social networks

## Abstract

Most sociophysics opinion dynamics simulations assume that contacts between agents lead to greater similarity of opinions, and that there is a tendency for agents having similar opinions to group together. These mechanisms result, in many types of models, in significant polarization, understood as separation between groups of agents having conflicting opinions. The addition of inflexible agents (zealots) or mechanisms, which drive conflicting opinions even further apart, only exacerbates these polarizing processes. Using a universal mathematical framework, formulated in the language of utility functions, we present novel simulation results. They combine polarizing tendencies with mechanisms potentially favoring diverse, non-polarized environments. The simulations are aimed at answering the following question: How can non-polarized systems exist in stable configurations? The framework enables easy introduction, and study, of the effects of external “pro-diversity”, and its contribution to the utility function. Specific examples presented in this paper include an extension of the classic square geometry Ising-like model, in which agents modify their opinions, and a dynamic scale-free network system with two different mechanisms promoting local diversity, where agents modify the structure of the connecting network while keeping their opinions stable. Despite the differences between these models, they show fundamental similarities in results in terms of the existence of low temperature, stable, locally and globally diverse states, i.e., states in which agents with differing opinions remain closely linked. While these results do not answer the socially relevant question of how to combat the growing polarization observed in many modern democratic societies, they open a path towards modeling polarization diminishing activities. These, in turn, could act as guidance for implementing actual depolarization social strategies.

## 1. Introduction

Agent-based models of social phenomena, using concepts and tools borrowed from physics (thus, sometimes dubbed sociophysics) are now a well-established research domain, involving not only physicists but also psychologists and sociologists. Within this field, studies of opinion dynamics are one of the subdomains which is enjoying rapid theoretical growth, and increased collaborations with empirical social studies and computer modeling. The models, especially when coupled with huge repositories of electronic data (describing various aspects of human behaviors), offer unprecedented capabilities of answering important questions and even predicting social phenomena (e.g., election results). However, this progress has a darker side as well.

As the present author noted in [1], there are serious ethical issues which should be considered when we develop “too accurate” models: they may be used outside the research domain, serving and guiding commercial companies, politicians, populist movements (such as the anti-vaccination activists), etc. We stand, thus, at the crossroads, deciding how to develop our research interests, so that its social benefits overcome the potential dangers of misuse [2].

As one should practice what one preaches, the current paper is an attempt to lay the groundwork for the study of how to combat the growing polarization observed in many democratic societies. This polarization not only diminishes our capacity for true deliberative dialog and for working out solutions to problems that would enjoy popular support, but also splits societies into conflicted parts, opens avenues for extremists, autocrats and manipulators, and threatens the very core of democracy.

Yet, while the amount of sociophysical research devoted to polarization itself is significant, the studies focused on reversing it are very few (e.g., [3]). This may be due to the fact that, for most of the many types of opinion dynamics models, the stable final states of the simulations are either a uniform consensus state or some variant of a polarized, divided one. So, we are faced with two non-desirable outcomes: uniformity without the chance for dissent or an irrecoverable split (often with the domination of extreme views). Since these results qualitatively correspond to empirical social observations, we (the modelers) often stop there, satisfied with our work, even if we oppose the actual social situation.

The goal of this paper is to present novel results of agent-based models which combine two basic mechanisms. The first is the polarization-leading tendency to prefer social connections with like-minded people, and to adjust one’s opinions to those of the in-group. This has already been modeled in many ways in the existing literature. The second mechanism is designed to represent the benefits resulting from the diversity of points of view, making local groups more effective and more resilient to external challenges. The benefits of diversity of experiences, viewpoints and competencies in solving problems are well documented in social sciences but have not been, as far as the present author knows, incorporated into opinion modeling approaches. We note that these novel mechanisms are difficult to incorporate in many traditional formulations of opinion dynamics ABMs **because they influence the opinions indirectly**. Rather than comparing one’s opinion with that of another person or group and adjusting it as a consequence of this comparison, the new mechanisms rely on the benefits of being in a diverse group when we face certain external challenges, such as disasters, economic hardship, conflicts, etc. The use of the utility function allows the combining of these direct and indirect approaches.

The new models presented in this paper are oversimplified on purpose. Rather than attempt to model the complexity of human societies and interactions in any degree necessary for quantitative descriptions or predictions we wanted to check if one can introduce, in a nontrivial way, anti-polarization or pro-diversity mechanisms into existing ABM opinion dynamics approaches. The answer turns out to be positive. We show that, for different model frameworks (ranging from a very simple 2D Ising-like system to a scale-free, dynamic network formulation) and for different postulated interaction models, **it is possible to reach a stable final system state, in which, locally and globally, the model exhibits significant diversity of connected agent states**. This result does not show the way to combat polarization and extremism, but it provides (hopefully) suggestions as to modeling of actions that could reverse the current divisive tendencies—without sacrificing plurality of views and opinions.

## 2. Social Sciences and Opinion Modeling Background

### 2.1. Polarization in Modern Societies

Polarization is seen as a growing threat to the modern world (at least the democratic part) [4,5]. It is observed in many societies [6], and is generally decried as something bad, even toxic [7]. The fact that many partisan media pundits and politicians put blame on “the other side” does not invalidate the real negative effects of polarization [8].

Still, what is actually meant by “polarization”? Some commentators focus on *extremization* of views, especially when the division becomes so strong that no consensus is possible. Another approach is to focus on the growing separation of modern, democratic societies into “tribes”. There are studies that confirm that political views are now a major factor in establishing personal relationships, even love. People refuse to have contact with those representing the other side. Studies confirm the subordination of opinions on specific subjects to the general partisan allegiance mentioned above. This increases the divisions between “us” and “them”. There are multiple “flavors” of polarization, among them *ideological polarization*, *affective polarization* [9] and *factual belief polarization* (for recent work, see, for example, [10,11,12]). There are other subcategories, focusing, for example, on psychological origins, such as *moral polarization* [13,14]. This plurality of definitions and meanings [15,16] sometimes leads to differences in research conclusions, as noted in [17]. Much of the research looks at the role of mass media and social networks in the growth of polarization [5,18,19,20,21]. Similarly, there are many psychological studies of the effects of exposure to differing views (and, conversely, to limiting contacts to like-minded “bubbles” and “echo chambers”) [22,23,24,25,26,27,28].

In the current work we focus on affective polarization, defined as the tendency to view opposing groups negatively, which leads to increased social separation between in-group and out-group members [29,30]. The actual political (moral, economic, social) issues and opinions are treated as subservient to ideological identity or mindset (which may be dictated or defined by partisanship).

### 2.2. Reducing Polarization

With the general perception of the negative effects of the current growth of polarization, it is no surprise that multiple lines of research are focused on decreasing the divisions. There is no room to review all the proposed strategies and analyses of their effectiveness. A sample of the recent literature is given by: [24,31,32,33,34,35,36,37,38,39,40,41,42,43]. The evidence is mixed, in that some depolarizing methods work in some environments, but fail in others. Many require specific conditions, which can be achieved in experimental settings (for example, they work for willing participants and in isolation from external influences). Very few proposed approaches promise the scalability required to influence large, already-divided groups or whole societies. Therefore, despite the importance of the subject, progress in the understanding of depolarization and in practical implementation of scientifically-based strategies to address it is limited (An important factor making practical depolarization difficult is that many politicians (either currently in power or in opposition) actively promote polarization, as the means of retaining their status and positions). There is, therefore, an opportunity for ABM approaches to add to the current state of knowledge.

### 2.3. Diversity Benefits

As we noted, most of the analyses of modern-day polarization discussions focus on the negative consequences of diveristy of opinions (although there are some mentions of positive effects, such as greater participation in public issues [17]). Additionally, there is a general perception that polarization (especially political polarization) increases, not only in the US (the most studied society), but in other countries as well. These observations lead to some fundamental questions.

With the many faces of polarization, we may ask, **“What exactly is the desired state of “unpolarized” or “depolarized” society”**? One of the answers is a state in which all actors hold uniform views. No division, no conflict, no diversity of opinions. However, such a state is clearly not too desirable for democracy. An alternative is a state in which those who hold diverse views are connected and able to interact (peacefully). Most likely, this is what journalists mean when they call for the return to a non-polarized state. To contradict this point, we recall here the arguments of Baldassari and Page [44], who noted that affective polarization leads to significant social segregation, extending into business, economic and social relations. This division results in a lack of diversity in local social clusters, which impacts ability to make sound decisions. Deliberative processes lacking sufficiently broad perspective are known to lead to group-think and importantly, in the context of polarization, to extremization of views. Thus, we use **the system in which the actors retain connectivity (enabling cooperation) with others not sharing their opinions as the working definition of the non-polarized state**.

The second question seems to be of a seemingly trivial nature: **If polarization is increasing, why was it smaller before**? What has changed or destabilized less-polarized social systems? There are several possible explanations (as is often the case in social sciences). For example, one could test whether the increase in polarization is the result of greater capacity to freely define our social environments, brought about by the recent ubiquity of electronic communication technologies. In the not so distant past, people had much less opportunity to choose their associates. The environments of the family, workplace and home neighborhood imposed contacts with people with potentially different views. Escaping such contacts was much more difficult than in modern, largely virtual, social environments. The necessity of finding a *modus vivendi* with people who did not share a person’s beliefs and opinions was even more important, because in the local environments mentioned above there was no possibility of using anonymity (available in the virtual world) as a shield.

Such an explanation, stressing the role of technological change, is certainly plausible, but it does not completely solve the issue. The preference to associate with like-minded people, the division of the world into in- and out-groups, the mechanisms for assuring the cohesion of the in-groups (through upbringing, social censoring, punishment, etc.) are universal and well-known in human history. Therefore, one may turn the question around and ask, “**Why are we not fully uniform in our ideological identities** (**at least locally**)”? Is there an evolutionary benefit to diversity of viewpoints and mindsets, or is it just a result of inevitable randomness, which cannot be fully mitigated by nurture and censure? What are the necessary conditions for such uniformity of mindsets and when does it happen?

Since a few decades ago, a whole branch of research has been devoted to studies of the benefits of viewpoint/mindset diversity. The two main identified benefits are related to group performance and resilience. Diversity of knowledge, experiences, mental models, and available tools allows groups of people to achieve better results in the creation of new solutions, with faster and better decision making. In facing problems or dangers, diverse groups are more resilient. Both effects have positive consequences for the groups: increased performance reaps direct benefits, while greater capacity to cope with adverse circumstances protects groups against dangers. These hypotheses have been studied and confirmed in social and economic contexts (see, for example, [45,46,47,48,49,50]), and recent work has also begun to address them in the ABM approach [51,52]. In the present approach, we simply assume that (local) diversity brings some sort of benefits, which may be expressed in the language of agent-based models. Such an assumption offers the chance to combine, in the same ABM framework, the benefits of diversity and the tendencies to segregate using similarity of viewpoints. This combination may offer crucial advances in ABM studies of the polarization growth and in potential countermeasures.

### 2.4. Volatile Opinions Versus Stable Mindsets

One of the most frequently used premises in studies of opinion dynamics is the assumption that individual opinions are easily changeable. In part, this approach results from the history of sociophysics and the exploitation of the analogy between social systems and magnetic phenomena. Comparing opinions (especially binary ones, characteristic for political support or yes/no decisions) to atomic spins made the rich instrumentarium of statistical physics easy to use.

However, in many social situations, we do not deal with opinions which are as easily changed as atomic spins. In reality, it is not easy to change someone’s opinion. The qualitative analogy with rapidly oscillating spins may be deeply misleading, mainly because of the difference in timescales of individual and global processes. In physics, there is a gap of many orders between characteristic times of individual spin flips and those of macroscale changes, such as sample magnetization, while in social situations individual and societal change timescales may be comparable).

Thus, in our approach, we propose to go beyond single, fast changing “spin-like” opinions, and to focus, instead, on much more stable “mindsets” or “viewpoints”. These mindsets may be attributed to various psychological origins. For example, one could use the Big Five traits of psychology (extroversion, agreeableness, openness, conscientiousness, and neuroticism) and try to understand someone’s behaviors (including opinions on specific issues, reactions to other people, or mass media) through the combination of traits characterizing this person. Alternatively, we can consider basic moral categories (harm/care; fairness/reciprocity; ingroup/loyalty; authority/respect; purity/sanctity) as the background of the individual mindsets [53,54,55,56,57].

Both of these foundations are known to determine general political stances (e.g., progressive/conservative) [58], which, in turn, define opinions on specific issues (abortion, gun access, immigration…) They are found (in slightly differing variants) in most cultures. They remain (relatively) stable for a given person for long periods. They determine behaviors, including personal decisions and social associations. In political terms, there are strong indications that certain mindsets are associated with conservative or liberal viewpoints and political party support [59]. As partisan loyalty, in many cases, surpasses fact-based reasoning on specific issues, a focus on slow-changing characteristics seems reasonable.

The moral foundations of personality are not the only ones proposed in the psychological literature. In his already cited *Diversity Bonus* [47], Page characterizes an individual “repertoire” of behaviors as being based on information, knowledge, heuristics or tools, representations and mental models and frameworks. He recognizes also that there are additional categories of principles and standards. While some of the components of the individual repertoire may change relatively fast (e.g., available and pertinent information), others, such as cumulative knowledge, tools, perspectives, and representations, as well as the mental models, are much more stable.

### 2.5. Opinion Dynamics Using Agent-Based Models and Physical Analogies

Developed since the 1980s, the field of modeling of opinion dynamics contains a large number of different models, which may be grouped into several “families”. Some focus on one-to-one interactions (one agent trying to convince the other, or two agents sharing a discussion and trying to convince each other, for example the voter model [60,61,62] or the bounded confidence model [63,64,65,66]). Other models use group influence approaches (where a suitably treated opinion of a group surrounding an agent influences its opinion), for example, Galam models [67,68,69,70,71], the social impact model [72,73,74,75,76] or the Hegselmann–Krause model [77]. For a review of the potential variants, see Castellano et al. [78].

While, initially, the models focused on the processes leading to consensus within social groups, later versions devoted much attention to polarization and transition to more extreme views. One way of achieving polarization is by assuming the existence of special classes of agents (inflexibles, zealots, or extremists, as in the following: [79,80,81,82,83,84,85,86,87,88,89]. If inflexible agents represent opinions at the extreme ends of the allowed spectrum, they may convince moderates, creating a strongly polarized society. In fact, one does not have to include any special class of inflexibles into the model. For example, within the bounded confidence framework, all that is required is that extreme views are associated with decreasing tolerances. In such a situation, most agents not only move to extreme opinions, but, at the same time, become inflexible [90].

Encounters between people in which they discuss their opinions often also involve the arguments used to support these opinions. When two persons share similar views, they can intensify their opinions by providing each other with new ways of supporting them, becoming more extreme as a result. This approach, called persuasive argument theory, has been used in several polarization models, for example, [91,92,93,94,95]. Other approaches were proposed in [96], in a modular, hierarchical approach and in [97], in which modeling effects of multiple issues are considered together.

In addition to the dynamics based on modification of changes of agents’ opinions, there are models considering co-evolution of opinions and social networks connecting the agents, for example, [98,99,100,101,102].

As one can see, there are multiple solutions proposed to resolve the problem of polarization, and, in general, the complexity of opinion dynamics. This wide variety of models reflects an even greater variety of explanations proposed in social sciences, often connected in complex ways or even mutually contradictory. Due to the social importance of the persistent conflicts and increasing divide spanning all levels of democratic societies on many issues, the current generation of opinion models often focus on the issue [103]. However, as already noted, the number of models devoted to reversing the polarization trends remains limited (see, for example, [34]).

## 3. Universal Agent-Based Model Framework for Co-Occurring Social Processes

The literature devoted to modeling of social opinion dynamics is today very rich, with multiple approaches describing individual behaviors in response to external stimuli, in models of the influences that agents may experience (from other agents or from mass media, social communication platforms etc.) and in different social network types (from simple/abstract to detailed copies of actual social interaction patterns).

The framework we present in the following is designed to combine at least a subset of these approaches within a single mathematical formulation, allowing flexible modifications. In particular, we aimed at a consistent language encompassing behaviors of the so-called “complex agents” (i.e., agents whose characteristics are multi-faceted and whose behavior may depend on the context, such as the environment they are currently in or their memories of past occurrences [104,105,106]). A similar complexity is allowed for external stimuli and influences, which also share psychological multidimensionality (e.g., combining information, persuasions, emotions), and for context dependence.

The proposed formulation is not a “cure all” and might be unsuitable for some social situations. After all, our knowledge of human behavior is far from perfect. Psychology and sociology contain many unresolved problems and propose conflicting hypotheses, or even empirical observations. However, we hope that our approach is interesting and useful, especially in regard to the goal of understanding the relations between mechanisms driving the opposite trends favoring uniformity and diversity in the same system.

### 3.1. Model Background and Definitions

We consider *N* agents interacting on a network (the agents are hereafter indexed by letters $i,i^{\prime }$…) The number of agents may be fixed (which is typical) or variable. Each agent is internally characterized by a *K*-dimensional “mindset” vector $A=\{A_{k},k=1,K\}$. The components $A_{k}$ can correspond to separate individual characteristics of the agent, such as the agent’s opinions on specific topics, knowledge, emotional state, moral foundational values, psychological traits, or even aspects such as social, marital status, wealth, etc. Depending on the choice of the modeled situation, the values of $A_{k}$ may take continuous or discrete values (e.g., ±1).

The dynamics of the system may incorporate several aspects. In response to circumstances, the agent might change the values of $A_{k}$. The agents, in the course of the simulations, can also rearrange the connecting network, for example, by deleting or creating links with other agents. These changes are achieved through maximization of the individual utility functions of the agents U(i). It is also possible to consider groups of agents acting together to maximize their overall utility function, rather than individuals. In what follows, we focus on the individual agent’s utility, so we drop the reference to the agent index *i* from the definitions of *U*.

The reason for formulating our model on the basis of the utility function is twofold. First, the utility function allows simple and direct analogy with physics. More importantly, it offers the capability to compare effects of different mechanisms driving system dynamics. Many existing models of opinion dynamics are based on specific formulae linking opinions before and after interactions in the system (for example the equations governing agent opinions in the Bounded Confidence models). Modification of such equations, through inclusion of new mechanisms, is often far from obvious, and sometimes impossible. Introducing the utility function components for various types of influences and interactions creates a common platform, allowing comparison of their effects—similarly to the various contributions to the energy of a physical system. In most cases, it is possible to re-formulate the traditional approaches (such as the Bounded Confidence models or social impact ones) in the language of the utility functions and Monte Carlo dynamics.

### 3.2. Utility Function Definitions

For a specific agent, the utility function is assumed to be the sum of two components. The first, $U_{I}$, describes the effects of interactions with neighboring agents. The second, $U_{E}$, describes the effects of external influences on the agent. Such a distinction suggests similarity to physical atomic systems.

We define the interaction part of the utility function as follows:(1)$$U_{I}=G\left(A,\left\{A_{n}\right\}\right),$$
where the function G depends on the strength of the interactions and a combination of characteristics of the agent (A) and of other agents with whom the individual agent interacts (the agent’s “neighborhood”), denoted collectively by $\left\{A_{n}\right\}$. Depending on the simulated system, the definition of the neighborhood may be different, ranging from the whole system (all agents interacting with each other) to only immediate neighbors surrounding the agent. Thanks to this definition, $U_{I}$ for a given agent may include a self-interaction component. In practical implementations, the neighborhood influences may be divided, for example, into parts, due to the nearest neighbors (NN), the next-nearest neighbors (NNN) etc. If the network connecting the agents is weighted, the form of G may include the connection strengths of the links. The formalism also allows for pairwise agent-to-agent interactions, when the function $G(A,\left\{A_{n}\right\})$ “uses” only a single agent from the neighborhood $\left\{A_{n}\right\}$.

The second part of the utility function describes the effects of external influences on the agents, such as the effects of mass media, cultural and legal circumstances, and the effects of natural and man-made disasters, etc. In the language of sociophysics, the most common example of such external influence is the analogy with the role of magnetic field in the Ising model. These external influences may be described as
(2)$$U_{E}=F\left(F,A,\{A_{n}^{\prime }\}\right).$$
Here, as before, A denotes the characteristics of the agent, $\left\{A_{n}^{\prime }\right\}$ denotes the characteristics of its neighborhood (which may be, in principle, different from the neighborhood used for the interaction part $\left\{A_{n}\right\}$), and the vector F denotes the strength of the external influence. We used a vector notation for F, as the influence may differ for different components of the agent characteristics, enumerated by *k*.

The social rationale for such a general form of the $U_{E}$ part of utility functions aims to reflect the human capacity to cooperate in response to external pressures or influences. Some existing opinion dynamics models have already considered selected external influences (e.g., opinions encountered in mass media), but, typically, the effects have only depended on the characteristics of the influenced agent A. In many situations, instead of reacting on his or her own, a person can elicit the support of associates (family, friends, tribe…). The response to a challenge might depend on a larger group or a single member who is best prepared to respond. We also learn to react from our peers. No man is an island in this context, and the collective, cooperative response may favor diversity.

This is the situation on which we focused in the present paper. **We attempted to formulate, in the utility function framework, the benefits of using group-based responses to external pressures or challenges. Such challenges might be, for example, posed by natural disasters, large scale conflict or economic situation, which demand agent activity. As already noted, the capacity to draw from a variety of experiences, viewpoints and competencies might positively influence the results of the chosen actions. Such positive contribution to the utility function, based on better resilience resulting from local diversity are the core of the novelty of the models presented in the following parts of the paper.**

Lastly, we note that a given system may be characterized by multiple external influences. Such a situation is easily incorporated into the model by allowing the external influence part of the utility function to be defined as
(3)$$U_{E}=F_{1}\left(F_{1},A,\left\{A_{n}^{\prime }1\right\}\right)+F_{2}\left(F_{2},A,\left\{A_{n}^{\prime }2\right\}\right)+\ldots$$
where the forms of functions $F_{1}$, $F_{2}$, fields $F_{1}$, $F_{2}$ and neighborhoods $\left\{A_{n1}^{\prime }\right\}$, $\left\{A_{n2}^{\prime }\right\}$ might be different.

### 3.3. System Dynamics

It is assumed that, during the simulations, the agents would perform actions aimed at increasing their overall utility function. The model considers two actions available to an agent at each step of the simulation:Change the agent opinion or viewpoint to increase the utility value. In the language of the framework presented above, this would be a change in the value of its characteristics A. For example, in the binary ±1 model, it would mean a switch of one of the components of A (similar to a spin-flip in a magnetism model). For models in which characteristics A take continuous values, the change may be more subtle and complex.Change of the network neighborhood of the agent. For example, cutting a link with one of the neighbors and establishing a new one with a different agent (link rewiring or migration). In the case of weighted networks, it may be more subtle; for example, a change in the weight of the link between two agents.
The two options presented above have the advantage of preserving the number of links and agents, making statistical comparisons straightforward. It is possible to imagine other behaviors of agents aimed at maximizing the utility function, for example, the growth or shrinking of the system size by creation or deletion of links or agents. In such a case, one would probably have to move from the individual utility measures to maximization of the averaged utility function.

In other words, in action 1 the agent adjusts his or her own characteristics to fit the environmental and external influences, while in action 2 the agent adjusts the environment (neighborhood) to fit his or her own characteristics. These actions may, of course, be combined together, creating a complex dynamics patterns. The two models presented below were chosen to examine the effects of the two types of agent actions **separately**, to simplify the analysis and to see if the pro-diversity component of the utility function affects the results of the two mechanisms.

## 4. Use Example: Extension of the Ising Model through Addition of Pro-Diversity Field

The general description of the model framework presented so far is complex, on purpose, to capture a wide range of potential psychological/social situations. It is, however, possible to reduce it to a situation readily comparable to the standard Ising model of magnetic phenomena. The Ising model is one of the most widely known theoretical applications of statistical physics, which, despite formal simplicity, exhibits complex behavior; in particular, the potential for a phase transition between a random, disordered state and a fully organized one. We note that, historically, the loose analogy between binary opinions and atomic spins, and the appearance of the ordered fully magnetized phase and general consensus of opinions was one of the first examples of sociophysical opinion dynamics models [72,107,108]. Such approaches equated binary opinions with atomic spins and the global (average) opinion with system magnetization. The analogy obviously has several weaknesses, namely, the following: spin fluctuation dynamics are very different from the individual opinion changes, and social interaction networks are different from lattice-based topologies used in standard Ising model formulations. There are also problems in mapping physical terms to social phenomena. While employing the analogy between sample magnetization and the overall opinion in the studied system is natural enough, there is no obvious social equivalent to physical temperature. For simulations, especially the Monte Carlo (MC) models, the “temperature” variable influences the relative frequencies of events, so it is a “technical” parameter. In physical applications of such simulations, the MC temperature can be equated with the thermodynamic one. However, for social simulations there is no obvious corresponding observable variable. Nevertheless, the notion of a transition from a random state to an ordered one was thought to be attractive enough for a qualitative opinion dynamics description. In what follows, we duly acknowledge these weaknesses, focusing on the potential to combine the self-ordering properties of the ferromagnetic Ising model with an additional diversity-enhancing mechanism. The goal was to see if such a mechanism might significantly impact the transition to a fully organized final state.

To start with, we considered only the first option of agent activity, i.e., change of the agent’s characteristics A, without changing the network (which is a simple 2D square lattice), as in the Ising model.

Then, we reduced the dimensions *K* and *L* to 1. Therefore, an agent was characterized by a single value *A* (we interchangeably used the named opinion and spin to emphasize the similarity of the contexts). If we assume that *A* can take ±1 values, we obtain the direct correspondence with the atom spin of the Ising model. However, in addition to reproducing the standard Ising model, our formulation allows its **expansion describing combined effects of magnetic field (*H*) and another type of external influence, which we call “pro-diversity field” (*F*)**. Below, we present the results of the simulations of the effects of such modifications.

The model was defined on a lattice in which each agent has exactly 4 neighbors. Agents’ activities are limited to the change of their viewpoints *A*, which are assumed to take values of ±1 (we sometimes use the notion of spin to describe these). The utility function is defined in “natural” units (in which the interaction strength for the Ising model *J* was set to 1), as follows:(4)$$U=U_{I}+U_{H}+U_{F},$$
where the first term is the interaction part of the utility function, and the $U_{H}$ and $U_{F}$ are two external field components. These parts are defined for each agent with spin *A*: (5)$$\begin{array}{ccc} U_{I} & = & A\sum_{n}A_{n} \end{array}$$(6)$$\begin{array}{ccc} U_{H} & = & HA \end{array}$$(7)$$\begin{array}{ccc} U_{F} & = & -F|A+\sum_{n}A_{n}| \end{array}$$ The first two terms are the standard ones known from the Ising model. In this, *H* is the magnetic field, and the index *n* denotes the four neighbors. The third term, which is the new introduction, is explained below. As the name of the field suggests, the pro-diversity field diminishes the utility function when neighboring agents’ opinions/spins are similar. This decrease is most significant (−5) for a fully aligned configuration and least significant (1) for configurations in which the five ±1 spins are divided into two to three (maximum local diversity).

The three components have different characteristics. Components $U_{I}$ and $U_{F}$ are symmetric with respect to the overall spin sum (magnetization) of the inclusive neighborhood and $U_{H}$ is antisymmetric. Both $U_{I}$ and $U_{H}$ “promote” uniformity, while $U_{F}$ penalizes it. The effects of these three contributions differ for each combination of the spin of the central agent and the sum of spins of its neighbors.

### Simulation Results for the Extended 2D Ising Model

Simulations were carried out using a NetLogo modeling framework [109]. The results presented here were obtained for a grid sized 101×101 (with periodic boundary conditions), which was large enough to minimize the finite-size effects and involved approximately 2000 Monte Carlo steps per agent, which waslong enough for the system to reach stable conditions.

Figure 1 presents simulations of magnetization (normalized sum of all agent spins *A*) as a function of temperature, for three values of *H*, and in the absence of the pro-diversity field. These results closely reproduced the standard Ising model simulations. The next Figure 2 presents similar results for the zero magnetic field *H* and for six values of the pro-diversity field *F*, which acts very differently from the magnetic field. Instead of the washing out of the phase transition (and increased range of temperatures with non-zero magnetization), increasing *F* shifted the transition from magnetized to non-magnetized state to lower temperatures. The same phenomenon was observed when in specific heat and magnetic susceptibility (Figure 3), both indicating the presence of a phase transition. One can estimate the dependence of the Curie temperature on *F* (shown in Figure 4). Value $T_{C}$ dropped rapidly with increasing *F* value.

Figure 5 presents the results of the combined magnetic field (H=0.5) and three values of the pro-diversity field (F=0.5,1.0 and 1.5). Even though it is relatively smaller, the magnetic field effect of spreading the phase transition to a wide range of temperatures dominated. Increasing *F* merely shifted the departure from the fully magnetized state to lower temperatures, preserving the gradual nature of the transition between the magnetized (uniform) and de-magnetized states.

In social language, these results might be treated as the expected results from the introduction of an influence that favors diverse environments extending the range of system parameters (in particular, the temperature) for which the system remains in a non-magnetized, “diversified” state. At the same time, the rapid decrease of $T_{C}$, which provides a quantitative estimate of the effect, is not easily predicted without simulations.

## 5. Network-Based Model of Competing Polarization and Pro-Diversity Mechanisms

The proposed general framework allowed for an arbitrarily complex description of agents, interactions, and external influences. At the low end of the available spectrum was the Ising-like model described in the previous section.

In the current section, we consider a more complex model. Its starting configuration was a scale-free Barabási-Albert network of agents, which is similar to actual networks identified in many social systems [110,111,112]. Such networks are characterized by significant diversity in agents’ connectivity, exhibiting power–law distribution. As noted previously, in this model we chose to limit the dynamics of the system to changes in the network structure, keeping the viewpoints of the agents fixed. The model focused, thus, on the opposite aspect to the one described in the previous section, where the connections on the square lattice were fixed but the individual viewpoints were allowed to evolve.

To model the system evolution we picked up a random agent, calculated the utility function and then attempted to rewire one of its connecting links to a new agent (keeping the overall number of links constant). The Monte Carlo procedure compares the original utility to the one calculated in the rewired condition, and either accepts the network change or rejects it. In the literature, such evolving networks are often described as dynamic. In the absence of the pro-diversity mechanisms, at low enough temperatures, the homophilic tendency of the agents leads to a separation of the initial single connected network into two or more separated parts. The goal of the simulation, including the pro-diversity components of the utility function, was to check if their presence might impact or reverse this separation of the system into disjointed polarized parts.

The agents were characterized by their viewpoints (fixed for each agent), with values taken from the range A∈[−5,5], denoting their political views (for example, in the US political context, the Democrat/Republican spectrum). The choice of the scale was purely arbitrary and chosen for programming convenience. We used two forms of the distribution of the mindsets: the first was a simple uniform distribution that spanned the available range. The second was a Gaussian distribution with a mean of zero and SD of 1 (limited to the [−5,5] range), which closely corresponded to a rescaled distribution of US political sympathies, calculated using data from [113,114]. The main difference between the two distributions was a much higher presence of extreme views in the uniform distribution, which increased the effects of conflicts and disagreements.

### 5.1. Utility Function

As in the case of the modified Ising model, our main interest was the comparison of the effects of interactions that lead to the grouping of agents sharing similar mindsets (i.e., leading to polarization), and the influences that drive increased diversity of social connections. The first part is described through the interaction part of the utility function $U_{I}$.
(8)$$U_{I}=A\left(\sum_{n}A_{n}\right),$$
where the sum is over the agent’s direct neighbors in the network, denoted by the index *n*. This utility function may be positive or negative. Positive values correspond to situations where the agent, and most of the agent’s neighbors, share the sign of the value of the political view, while negative values correspond to situations where the agents “disagree”. As a result, $U_{I}$ drives the system into polarization, understood as a separation of agents holding opposite views into disjoint subnetworks.

An important difference from the Ising model should be noted here: as the values of agent viewpoints *A* were taken from a continuous distribution, $U_{I}$ was close to zero for agents with A≈0, regardless of their environment (agents with “neutral” mindsets not “reacting” to the diversity/uniformity of viewpoints of agents around them). This is absent in the original binary Ising model, where the absolute value of spin is fixed.

The external part of the utility function was much more difficult to choose. There are multiple possible options to promote opinion differences between agents via a contribution to the utility function. Our aim was to use a formula that would not only **positively reflect the presence of variability** in the neighborhood of the agent, but also **scale with increased number of neighbors**. The latter condition is necessary, so that $U_{E}$ would not be overwhelmed by $U_{I}$ for agents with many connections. With this in mind, we decided to use two forms of the $U_{E}$. The first (“*absolute average*”) is the same as the one introduced in the 2D lattice toy model (Equation (Equation 7)):(9)$$U_{E1}=-F\left|A+\sum_{n}A_{n}\right|,$$
in which the more similar the viewpoints in the inclusive environment of an agent, the lower (negative) the utility.

For the second version of $U_{E}$, we used, as a measure of diversity, the product of variance of the mindset distribution in the inclusive neighborhood and the number of agents in this neighborhood (“*variance-based*”):(10)$$U_{E2}=F\left({\left(A-\bar{A}\right)}^{2}+\sum_{n}{\left(A_{n}-\bar{A}\right)}^{2}\right),$$
where $\bar{A}$ is the average mindset of the inclusive environment of the agent. Here, the utility is positive and increases with increasing diversity of agents in the neighborhood. The reason we multiply the variance by the number of neighbors is to preserve the same scaling with the agent connectivity as for the interaction part of the utility function.

In both cases, *F* is the strength of the pro-diversity field, a key variable of the model, defining the relative strengths between the interaction part and the pro-diversity one.

To facilitate navigation between the four possible combinations of model types, we provide a short summary in Table 1. Each simulation is described by one of the four combinations and the values of temperature *T* and the strength of the pro-diversity field *F*.

### 5.2. Agent Activities

As already noted, in this model we preserved the characteristics of the agents. The only option available was to rewire the connections. To focus on the polarization (segregation) and diversity-mixing effects, the simulations preserved the total number of links. Furthermore, the algorithm used prohibited the separation of a single agent from others. Each agent had to have at least one neighbor.

The simulations lasted for at least 1500 Monte Carlo steps per agent, which was sufficient to reach a stable value of the total utility function (which could be a global or a local maximum). The size of the system was set at 1000 agents, and the initial BA scale-free network generation process assumed 2 links per agent, on average.

In what follows, we distinguish the results from the choice of the external utility function ($U_{E1}$–absolute average or $U_{E2}$–variance-based); and the distribution of the agents’ characteristics *A*: uniform or Gaussian (the latter based on the actual estimates of liberal/conservative viewpoints in the US, scaled to the range of −5 to +5). Thus, there were four combinations of the model conditions, with the main variables being the strength of the pro-diversity field *F* and the temperature. Taking into account the inherent randomness of the network creation algorithm and of the Monte Carlo simulations, we ran at least 12 simulation runs for each combination of the model parameters. This allowed us to observe the statistical distribution of the final results, where the differences between individual simulations ranged from negligible to significant, depending on the specific configuration.

## 6. Results for the Network Model

As in the case of the extended Ising model, the simulations were programmed using the NetLogo language.

### 6.1. System Behavior Dependence on Pro-Diversity Field *F*

As the agents’ characteristics remained unchanged during the simulations (in contrast to the Ising model), in our analyses we focused on the network properties which might be used to measure local diversity. The first of such measures was the number of “contentious” links connecting agents with different opinions, shown in Figure 6. Specifically, we recorded the number of links between agents *i* and *j*, for which the absolute value of the difference $|A_{i}-A_{j}|$ was greater than 5 (for the uniform distribution *A*) or 3 (for the Gaussian one). This difference in thresholds was motivated by a much smaller number of agents holding extreme (|A|>3) positions in the Gaussian distribution model. Since we were primarily interested in the possibility of achieving stable diversity in our system, most of the simulations used a low temperature value for the MC process (0.05 in absolute units).

In all four studied cases ($U_{E1}$ or $U_{E2}$; uniform or Gaussian distribution), the general shape of the dependence of the number of contentious links on *F* was similar and, below a certain threshold (depending on the combination of conditions), it remained close to zero. Above this threshold, the number of such links increased to reach a plateau, with the final number depending on the combination of model conditions.

An alternative approach to measuring polarization/diversity in the system is based on the network segregation. Here, we used the sizes of the largest separate components into which the initial (fully connected) network configuration might split after the simulation, shown in Figure 7.

Again, the four studied cases shared similar behavior. At low values of the pro-diversity field *F* one observed the existence of two similarly sized components, into which the system was divided (plus a varying number of very small, typically two-agent, components separated from the network). Inspection of these two largest components showed that they were populated by agents with opposite opinions. So, for low values of the pro-diversity field, we recreated the separation of the system typical in strong polarization.

Above a certain threshold value of *F* (quantitatively coincident with the value for the number of contentious links), the final configuration changed dramatically. It consisted of a single largest component (in which agents with any opinion were present) and a larger number of very small, detached, components. In simulations employing the absolute average form of the pro-diversity utility function ($U_{E1}$), the growth of the largest component with growing *F* was rapid but still smooth. In the variance-based model ($U_{E2}$), the transition was step-like and might occur in a range of values of *F*.

Figure 8 presents the final network configuration examples for the two situations, below and above the threshold value of $F_{D}$. In the first case, we see a clear separation of the system into two parts, each grouping of agents sharing positive or negative values of viewpoint *A*. Above the threshold $T_{D}$, the final state consisted of a single largest component, combining agents of various mindsets, plus a number of isolated, small, or very small, components. The general network structure above and below $T_{D}$ was similar for both variants of the pro-diversity utility and for the uniform and Gaussian initial viewpoint distributions.

An interesting case was presented by the situation in which the starting point was not a random agent/link configuration with all agents connected in one component, but a polarized state, such as the one shown in the left panel of Figure 8. In other words, to the question, “Can we depolarize an already polarized system?”, the answer within the proposed model is yes.

Even at very low temperatures, when we increased the pro-diversity field above the threshold value $F_{D}$, the system evolved to the non-polarized state. This situation is presented in Figure 9. The figure presents a graphical representation of the network and the histogram of the absolute values of the differences of mindsets of agents connected by links for three situations: the starting (random) conditions of the simulation; the polarized state (at F=0.2) after 1500 MC steps per agent; and the depolarized state after the subsequent 1500 MC steps per agent using F=2. This final state was statistically very similar to the state achieved directly from the initial random conditions at the same F=2 value (right panel of Figure 8). This meant that the separation observed at low *F* was not irreversible, if one increased *F* sufficiently. In other words, increasing the importance of the pro-diversity field could actually de-polarize the system.

### 6.2. Dependence on the Temperature

For the model parameters for which a clear transition from polarized (separated) sub-populations to a dominant, single cluster was observed, we might, additionally, study the dependence on the temperature (as used in the Monte Carlo simulations).

Figure 10 presents the number of agents in the two largest connected components in the final configurations as a function of temperature. We considered two cases: the absence of the pro-diversity field and F=0.1, and a value below the threshold needed to reach one connected state. Simulations were carried out for the uniform distribution of the agent’s viewpoint and the definition based on the variance of the local diversity $U_{E2}$; corresponding to the upper left panel of Figure 7. Increasing the simulation temperature led to a transition from the separated (polarized) state to the connected final state. However, there was no single specific temperature at which the transition occurred. For every simulation, it might be different: there was a range of temperatures associated with the transition, centered around the average value $T_{C}$, which depended on the choice of the model parameters, and, in particular, the strength of the pro-diversity field. $T_{C}$ was defined via fitting a logistic distribution to the ratio of simulations ending in de-polarized final states.

The dependence of $T_{C}$ on *F* is shown in Figure 11, for the two cases of the pro-diversity field type $U_{E1}$ and $U_{E2}$, and the two types of the initial viewpoint distribution. Despite the fact that individual simulations led to different transition temperatures (with significant ranges shown by the error bars), the results were qualitatively similar. The average values $T_{C}$ decreased smoothly with increasing *F* as it approached $F_{D}$.

The results showed that both versions of pro-diversity field might lead to a situation in which the system might be largely non-polarized at relatively low temperatures; that is, contain one large connected component, composed of agents with differing viewpoints. Above a certain threshold value $F_{D}$ of the strength of the pro-diversity field *F*, the number of links between agents with different characteristics increased, which meant that both local and global diversity were preserved. For a fixed value of *F*, increasing the temperature led to a similar transition. Thus, while operationally the temperature and the pro-diversity field act in very different ways, they lead to similar final outcomes, wherein a large part of the system contained connected, diverse agents. The key difference in the final states created through the two mechanisms was that for high temperatures many links were dynamically rewired all the time, while for high *F* the final configuration might be “frozen” in the diversified state (for low temperatures).

## 7. Conclusions and Future Directions

In the current article, we introduced an expansion of the range of models of opinion dynamics, through the introduction of mechanisms that promote local diversity. We studied in detail two scenarios. The first might be described as an extension of the classical 2D square-lattice Ising model, with opinions limited to binary ±1 values and ferromagnetic interactions. The addition of a pro-diversity term (Equation (Equation 7)) changed the behavior of the system, both in the presence of a magnetic field and in its absence.

The second model considered a system designed to correspond somewhat better to social contexts. It assumed a dynamic network of agents, starting from a scale-free configuration. The agents’ characteristics, taken from continuous distributions, were assumed to be fixed, and the system evolution consisted of changes in the inter-agent network through link rewiring (preserving the total number of links).

Despite the fundamental difference in the nature of the changeable component of the system description between the 2D extended Ising model (Section 4), where we allowed changes in agents’ viewpoints but kept the neighborhoods fixed, and the Ising-like network model, presented in Section 5 and Section 6, where viewpoints were fixed and the neighborhoods changeable, the effects of the pro-diversity field were remarkably similar. In both models, a sufficiently high *F* led to a stable depolarized system, and the higher the *F*, the lower the temperature at which these transitions occurred.

Translating the simulation results into the language of opinion dynamics and positioning them in the original goal of combating excessive polarization, we admit that these are but first small steps. As noted in the introduction, one could treat the current model as a step backwards from more advanced and complex models already formulated, such as the Emotion–Information–Opinion model originally proposed in [115] (which was used to predict the results of the Polish elections in 2015 [116]) or models including biases and complex behaviors [105], which offer more depth in modeling the complexities of human behavior. The assumptions and results presented here are obviously oversimplified and qualitative. However, we hope that they open the path for future work with more direct applications. For example, we plan to study the effects of local problem solving activities and discussions involving varied viewpoints in controlled settings, considered by some to be the key to diminishing polarization (see works cited in Section 2.2, as well as the recent paper by Conrad and Lundberg [117]). How often, and how ubiquitous, should such activities be to have a significant impact on larger groups and societies? What is necessary to inhibit such disputes from turning into even more polarizing quarrels? The proposed framework easily incorporates such temporary and local (non-uniform) pro-diversity field models. Although the results presented here were based on a uniform, global, pro-diversity field formulation, the finding that increasing this field, even in a fully polarized society, leads to depolarization (see Figure 9) is very encouraging. This is because, although the current model uses a global and uniform field *F*, the actual dynamics are localized around individual agents. Therefore, a more realistic model, most likely focusing on mesoscale phenomena and nonuniform, time-varying *F*, seems very promising.

One more possible (pr even necessary) extension of the proposed model would be the inclusion of emotions into the utility function framework. Emotions, especially negative ones (fear, loathing, hatred), can easily trump rational considerations of risks and benefits, as parts of choices and actions. Therefore, their impact on *U* would likely be non-linear.

Another direction of research would be an attempt to model the effects of a danger that encompasses a whole society, such as famine or war. Recent examples of societies reacting to such crises range from a transition from fragmented society to a closely knit one in Ukraine, to civil wars in countries hit by drought or other catastrophes. Can we model these situations and distil the key factors differentiating the societal responses? This would correspond to one of the goals of Agent-Based Models, as set out by Epstein [118], namely, providing a guide for empirical social science studies. As noted recently in [119], polarization studies need a truly multidisciplinary approach if we are to understand the phenomenon and combat its excesses.

## Figures and Tables

**Figure 1 entropy-25-00568-f001:**
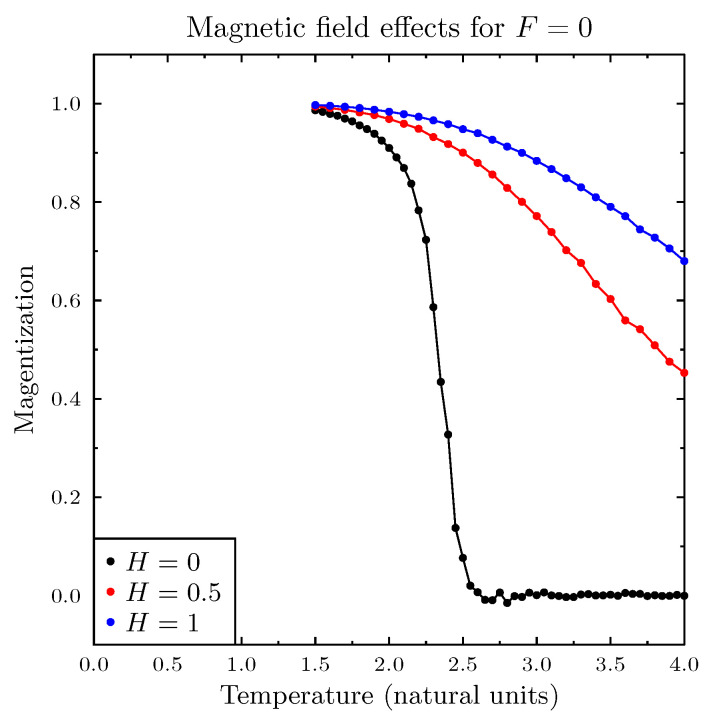
Magnetization as a function of temperature for three values of magnetic field *H*. The simulations reproduced results for the standard Ising model.

**Figure 2 entropy-25-00568-f002:**
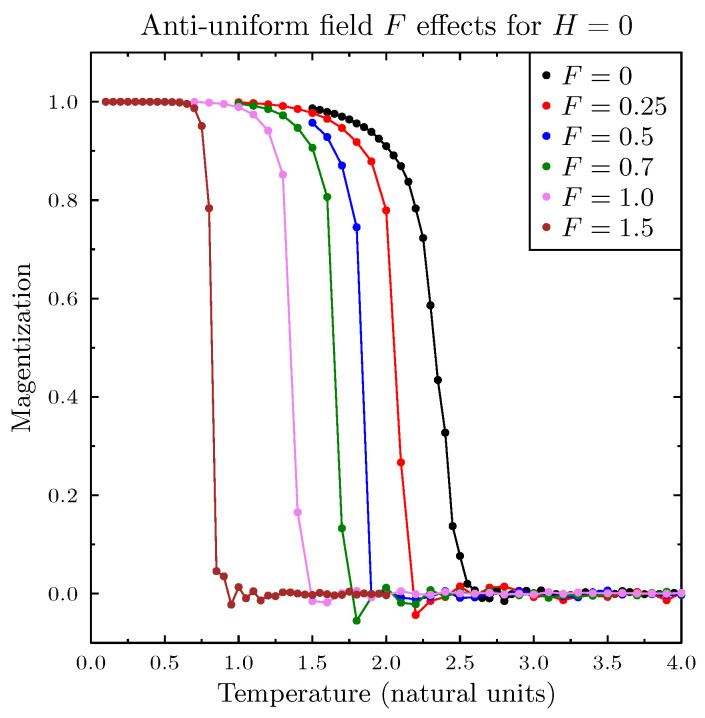
Magnetization as a function of temperature for several values of the pro-diversity field *F* and for zero magnetic field H=0. Increasing the pro-diversity field *F* shifted the Curie temperature $T_{C}$ to lower values (see Figure 4).

**Figure 3 entropy-25-00568-f003:**
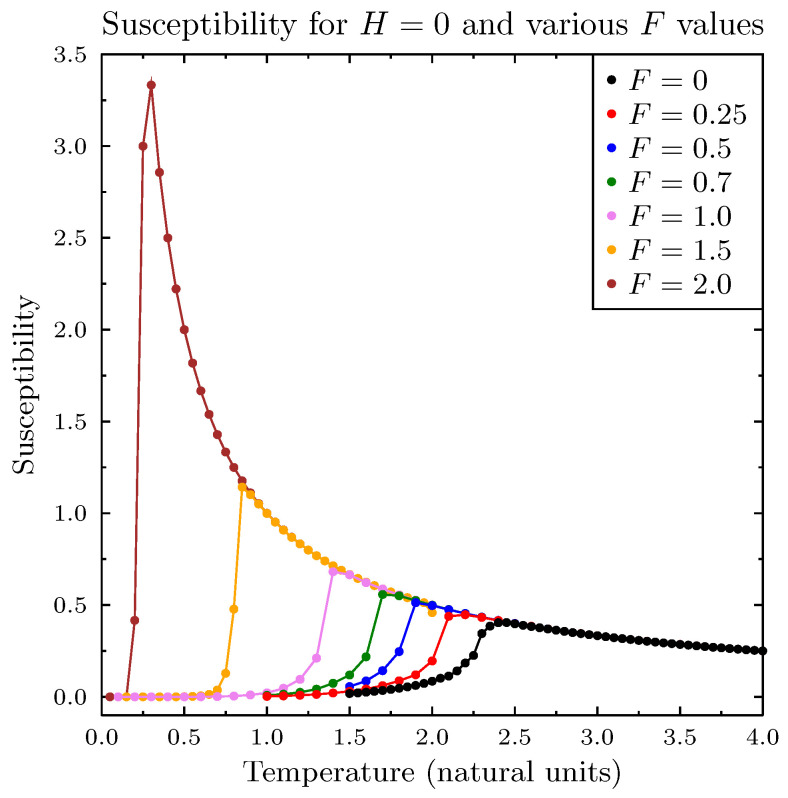
Susceptibility as a function of temperature for selected values of pro-diversity field *F* and for zero magnetic field H=0. Increasing field *F* shifted the phase transition to lower values. The transition became sharper when increasing *F*.

**Figure 4 entropy-25-00568-f004:**
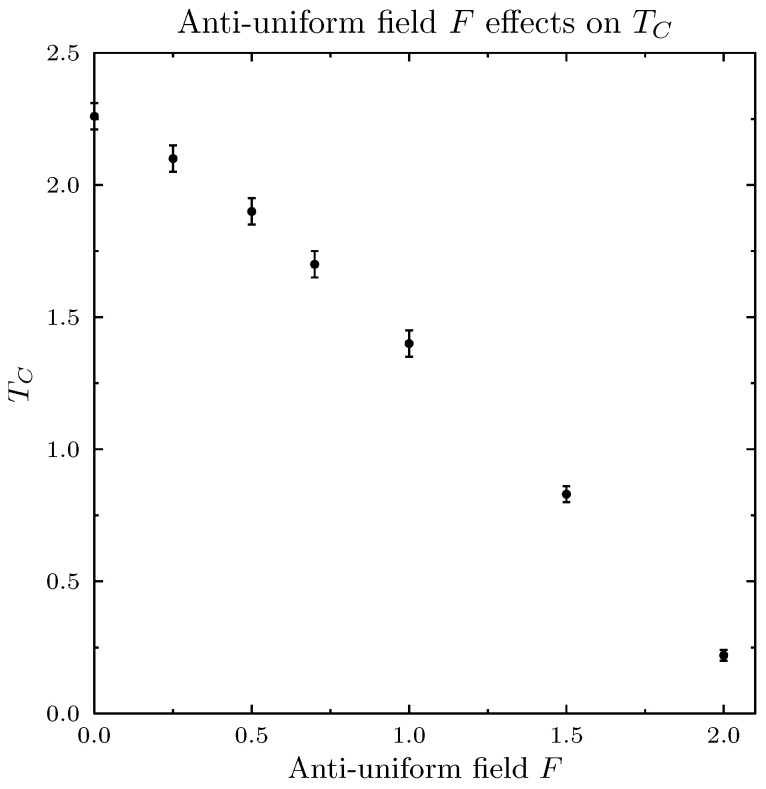
Curie temperature $T_{C}$ as a function of the pro-diversity field *F* for zero magnetic field H=0. Increasing field *F* shifted the Curie temperature to lower values. As temperature determines the ratio of the individual spin-flips, low values of $T_{C}$ mean that the system is globally depolarized with less individual “volatility”.

**Figure 5 entropy-25-00568-f005:**
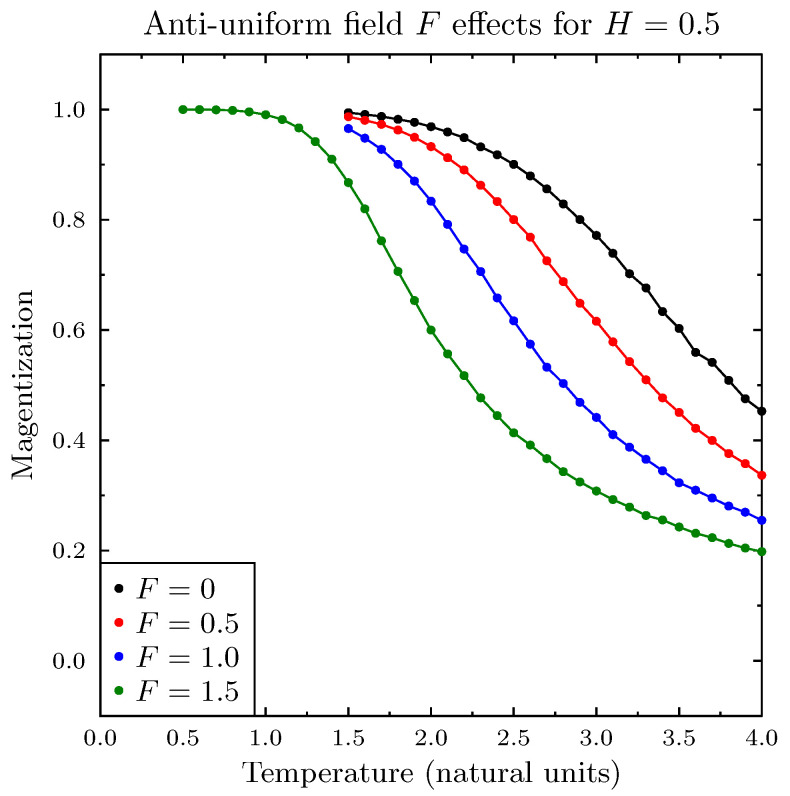
Magnetization as a function of temperature for several values of the pro-diversity field *F* for non-zero magnetic field H=0.5. There was no sharp transition, but the temperatures at which magnetization dropped shifted to lower values with increasing *F*. The error bars indicate differences between multiple simulations run for the same parameter values.

**Figure 6 entropy-25-00568-f006:**
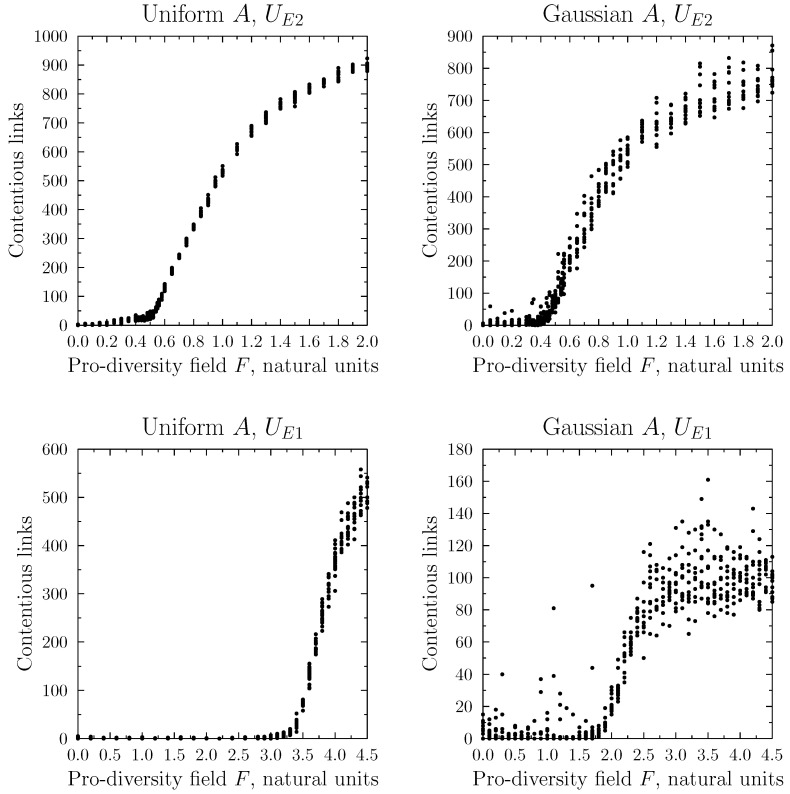
The number of contentious links between agents in the final configuration as a function of the pro-diversity field *F*. These “contentious links” are defined as links between agents whose mindsets differed by more than 5 (for the uniform initial distribution case) or 3 (for the Gaussian initial distribution). We recall here that the total number of links was 2000. For all simulations, the Monte Carlo temperature was set at a low value of 0.05. The four panels correspond to four simulation conditions, $U_{E1}$ (Formula (Equation 9)), $U_{E2}$ (Equation (Equation 10)), and the two initial distributions of the characteristics of agents *A*, uniform and Gaussian.

**Figure 7 entropy-25-00568-f007:**
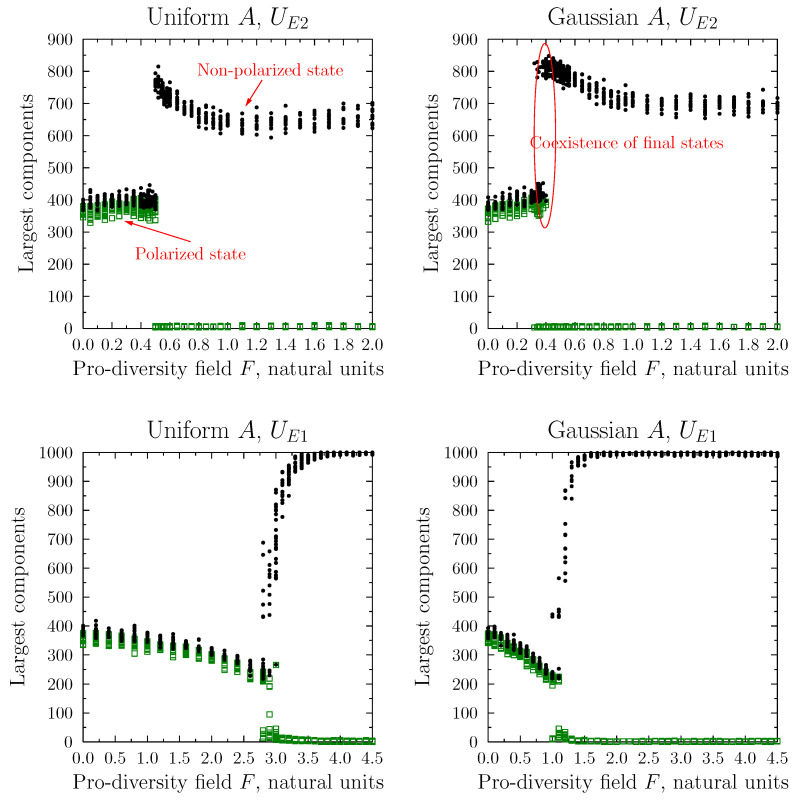
The size of the two largest connected components in the final network configuration as a function of the pro-diversity field *F*. The notation and conditions were the same as in Figure 6. Black circles denote the final sizes of the largest connected component; open squares (green) show the size of the second largest one. In the polarized state (small *F*) the sizes of the two largest components were similar, with many much smaller ones also present. Combining this result with the one in Figure 8 one can see that the two groups linked agents with similar opinions, creating a polarized state. Above a certain value, $F_{D}$, the simulations ended in a single dominant connected component, grouping agents with diverse viewpoints. For the absolute average version of the pro-diversity formula ($U_{E1}$), this component included most of the agents, while for the variance-based variant ($U_{E2}$), some 20–30% of the agents remained in disconnected, very small, groups. The transition between the two regimes (polarized and non-polarized) was very rapid, but there existed a range of *F* values in which a given simulation might result in either of the final states (the co-existence range being *F*).

**Figure 8 entropy-25-00568-f008:**
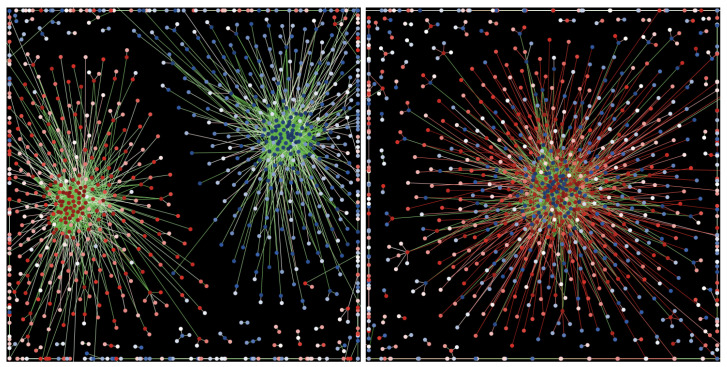
An example of the final network configuration for the simulations using the uniform agent mindset distribution and $U_{E2}$ pro-diversity field definition. The panels show final configurations for two values of the pro-diversity field F=0.2 (below the threshold $T_{D}$ at which agents formed one large cluster) and F=2 (above the threshold). The colors of the nodes correspond to the mindsets of the agents, from −5 (red) to +5 (blue), with the intensity of the color corresponding to the absolute value (so that white nodes represent A≈0). The color of the links corresponds to the absolute difference between the mindsets of the connected nodes: green denotes differences smaller than 3, and red larger than 3. Similar results are obtained for other variants of the simulations ($U_{E1}$, uniform/Gaussian distributions.)

**Figure 9 entropy-25-00568-f009:**
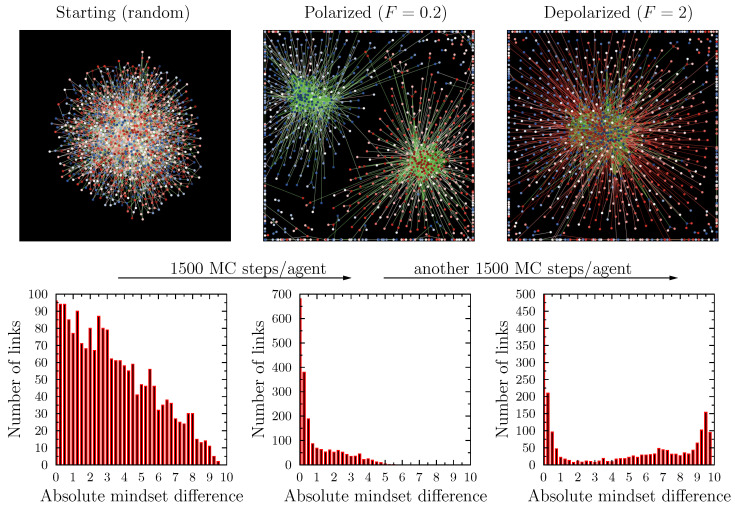
An example of the depolarization process due to the pro-diversity field. The columns show the network configuration and the histogram of the absolute value of the mindset difference between agents connected by links for three states in a single simulation (corresponding to model parameters, as in Figure 8). The left column shows the starting, random configuration. The middle column shows the polarized system (after 1500 MC steps per agent) obtained using F=0.2. The right column shows the system after another 1500 MC steps per agent, but this time with F=2. This final system was statistically identical to the one obtained by going directly from the initial random configuration. One can therefore state that an adequately strong pro-diversity field has the capacity to reverse the polarization due to homophobia.

**Figure 10 entropy-25-00568-f010:**
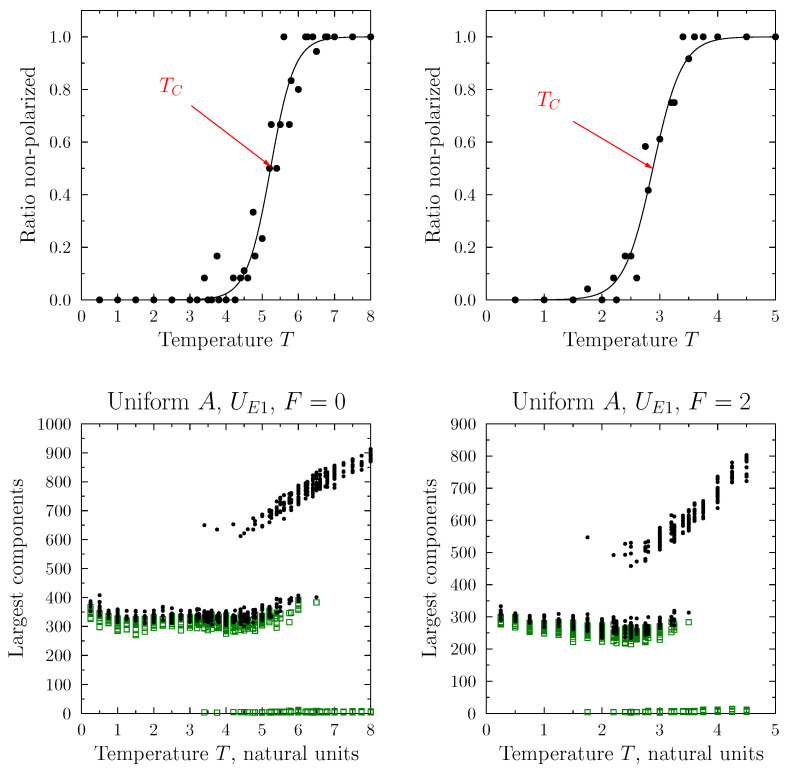
Sizes of the two largest connected components in the final state as a function of temperature. Simulation conditions: the absolute average pro-diversity ($U_{E1}$) formula and uniform initial viewpoint distribution. Bottom panels: size of the two largest connected components for individual simulations; for two values of the pro-diversity field strength F=0 and F=2. Top panels: ratio of simulations ending in depolarized state. The critical temperature $T_{C}$ was defined at the middle of the transition, the spread of the transition region was treated as an error in $T_{C}$ evaluation. Similar behavior was observed for other model variations (where $U_{E1}$ was replaced by $U_{E2}$, and the uniform distribution was replaced by the Gaussian one.) The lines in the upper panels were best fits using the logistic function, defining the average transition temperature $T_{C}$ and the associated error (see Figure 11).

**Figure 11 entropy-25-00568-f011:**
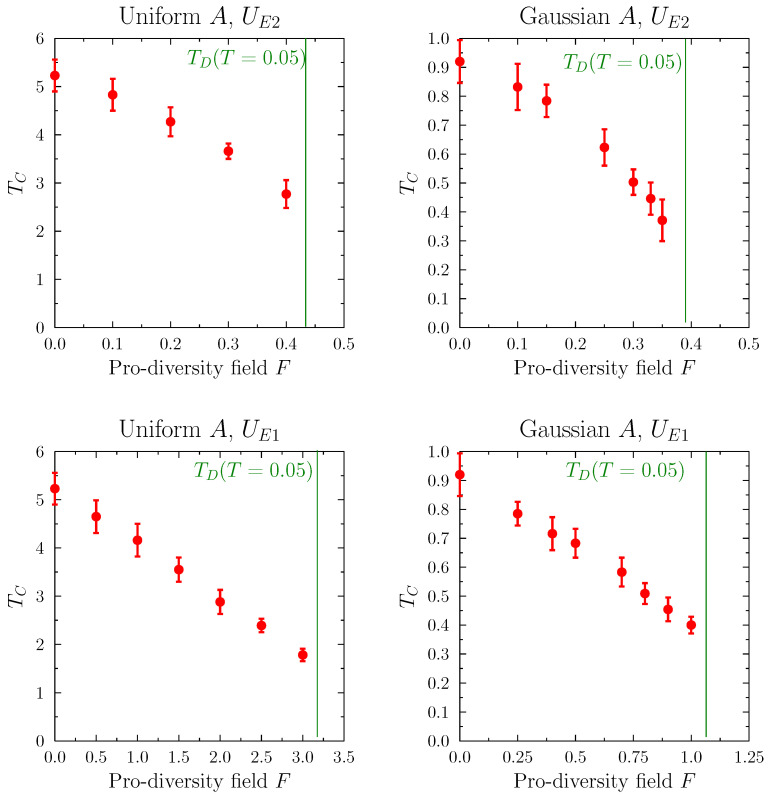
Dependence of the critical temperature $T_{C}$ on the strength of the pro-diversity field for the four cases, where the inter-agent part of the utility function was Ising-like. $T_{C}$ and its errors were calculated using the logistic function fits to the distribution of individual simulation results. Green lines indicate the approximate values of the pro-diversity field $F_{D}$ at which system transitions to depolarized state were at very low temperature (T=0.05). Despite the quantitative differences in $T_{C}$ and $F_{D}$ in the four cases, the behavior was qualitatively similar, in that the presence of the pro-diversity field lowered the temperature at which the final configuration was non-polarized.

**Table 1 entropy-25-00568-t001:** List of variants of the simulations for the network based model.

Pro-Diversity Field Formula		
Variant description	Acronym	Defining equation
Absolute average	$U_{E1}$	Equation (Equation 9)
Variance based	$U_{E2}$	Equation (Equation 10)
**Initial distribution of viewpoints**		
Uniform	random, uniform between −5 and +5	
Gaussian	random, Gaussian, centered at 0, with σ=1	

## Data Availability

The models used in this work are available from the Author upon reasonable request.

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
