# Peer review of "Social Depolarization and Diversity of Opinions—Unified ABM Framework"

_entropy, 2023, doi:10.3390/e25040568_

Round 1
Reviewer 1 Report
This is an outstanding piece, of wide interest and worthy of publication almost as-is. The emphasis on de-polarization is, as noted, fairly rare in the literature, and much needed. The study is thorough and well documented.
It is true that ‘benefits of diversity’ is simply dropped in as a parameter, but for a study at this stage that is both expected and acceptable.
All I have to offer is a few references beyond the extensive bibliography already provided here.
On senses of polarization, mentioned early in the paper:
Bramson, et al, “Understanding Polarization: Meanings, Measures, and Model Evaluation Philosophy of Science 84 (2017), 115-159.
Bramson, et. al, “Disambiguation of Social Polarization Concepts and Measures” (Journal of Mathematical Sociology 40 (2016), 80-111.
My apologies for the delay in my refereeing duties. This is an excellent paper that deserved better.
Author Response
We thank the Referee for the positive comments. Indeed, at the present stage, we simply added an assumption that (local) diversity has benefits, which may be described by an increase in the utility function. The origins of such benefits and the circumstances in which they become important are left for future studies, hopefully combining detailed social science empirical approaches and modeling. A short acknowledgment of this limitation has been added (lines 160-162).
The two references were added to the bibliography.
Reviewer 2 Report
This is an interesting paper that certainly can deserve publication. I however suggest various improvements before the paper can be accepted.
- The author should make clear whether the paper is a research paper or a review paper and structure it according to his choice. In the former case the author should explicitly describe the novelty and relate it to existing literature. Are the simulation and results new? In latter case the author should give more wide vision of the field. There are other approaches of methods of Statistical Physics to describe opinion dynamics and opinion formation – the kinetic theory approach. But they are even not mentioned by the author. There is huge literature, e.g. (and references therein).
L. Pareschi and G. Toscani, Interacting Multiagent Systems. Kinetic equations and Monte Carlo methods, Oxford Univ. Press, Oxford, 2014.
M. Lachowicz, H. LeszczyÅ„ski, E. Puźniakowska–GaÅ‚uch, Diffusive and anti-diffusive behavior for kinetic models of opinion dynamics, Symmetry, 2019, 11, 1024.
- I have a problem with mathematical formalism in Section 3.2. First of all, it is so general that it can mean everything (or nothing). Additionally, the notation system is not consistent – for example – Eq. (2) versus Eq. (3). If the function on the right-hand-side of Eq. (3) is a function of two variables then the product of these variables is not a linear function (it is a bilinear function) as it is written in line 291. Of course it is easy to understand the intention of the author. But the problem is that these equation do not have any deeper sense – they are just the formal play of symbols. It this “a novel universal mathematical framework for models…”?
- The author uses the notions from Statistical Physics related to the Ising Model. They usually have a clear physical meaning and are measurable. The situation is less clear in the case of Social Sciences. How can one describe “magnetic phenomena”, “temperature” in that context?
- If the intention of the author is to address his paper also to those that are not specialist in the Ising Model then I suggest to give more details related to the direct formulation of the model.
- The modelling in Section 5 is rather mysterious. Why the political choices are referred to interval [-5,5]? What are the details of the approach? I really suggest to revise this part of the paper and make the life of a reader easier.
Author Response
Ad 1. While the paper contains a rather extensive list of references it does not aim to be a review of the field. The main goal is to introduce and argue for a new direction of the ABM studies of opinion dynamics, namely introducing the effects of diversity in the modeled social systems and their role in combating polarization. The simulations, while certainly simplistic, are, to the best of our knowledge new in the field. While we recognize that sociophysical descriptions of social phenomena contain approaches that we have not mentioned, it is beyond our capacity to provide a comprehensive list of the multiple methods.
Ad 2. We than for drawing the attention to the lack of consistency: we have updated the equations to improve consistency. As for the “novel universal framework” – indeed the statement may be considered too bold. However, the current ABMs of opinion dynamics contain multiple formulations, typically linking opinions of agents at time t to changed opinions at time t+1 – which are often impossible to combine. The use of the utility function and MC approach allows much more flexible mixing of various mechanisms. A paragraph explaining this rationale was added (lines 280-291)
Ad 3 & 4. We have added a brief explanation of the Ising model and its history within social models (lines 345-361). We fully agree with some comments of the referee: while equating global opinion with magnetization seems easy enough, the equivalent of the temperature is by no means obvious (attempts to explain it in the literature are definitely lacking). In fact, in our opinion, the models of opinion dynamics have to take into account that individual opinion change characteristic times for different people (and agents) may span a large range of times, sometimes much shorter and sometimes much longer than the times of social opinion changes. This interplay is absent in most of the physical systems used as analogies – an interesting challenge for the modeling community.
Ad 5. The choice of the interval was purely arbitrary (made for programming convenience). A short explanation of this fact has been added to the text. On the other hand, the Gaussian distribution used for the realistic model parameters were pre-calculated to correspond to the actual political sympathies in the US society.
Reviewer 3 Report
It's hard for me to judge this paper--I lost track after about p. 8. The problem is that I am coming from the public opinion side, so I'm not familiar with the models and terminology. As far as the substance, I'll defer to other reviewers. It may be fine in its own terms, but its impact would be greater if it were more accessible to people like me. In order to make that possible, the authors would need to have more explanation of what the different parts of their model correspond to in terms of opinions. They make a start on p. 6 and p. 7, but their remarks are pretty general. I am looking for something more specific--that is, what does a "challenge" mean (is it just an effort to change opinions?), and exactly how the response would depend on associates. This kind of explananation is particularly important for the "pro-diversity field" given its importance in the simulations.
Author Response
We deeply appreciate the comments – coming from the outside of sociophysics. The paper is technically oriented, even though the motivation is interdisciplinary. Indeed, it is our ultimate goal to create a model which would not only be accessible to social scientists and public opinion researchers – but also provide some positive contribution to all people interested in bridging the gaps in our polarized societies. But such a model/application is still in the future.
The goal of the present paper is more humble: to test if the new approach proposed here can lead to depolarization. The testing is done using systems that are oversimplified on purpose, and which, as a result, are far from social realities.
Nevertheless, in the current version, we have expanded the abstract, introduction, the rationale for using the utility function (lines 324-341 and 364-369), and the opening of Section 5, which, we hope, provide „human readable” descriptions of the systems chosen for the simulations and why we chose them.
We also explain what we mean by „external challenges” and their indirect role in opinion distribution (lines 51-66).
Reviewer 4 Report
This is a concise review written after the request to accelerate the process initially planned to be completed by 15 March 2023.
It presents only an overall opinion.
The paper is innovative, logical, and formally correct, well-grounded in the existing body of knowledge.
Its main weakness, similarly as in many applications of models of collective phenomena, is typical.
At present, using any more or less advanced description of any collectivity, it is possible to call it "society" and then, using a more or less sophisticated model, obtain the results reflecting “new” characteristics of such societies. It is a kind of signum temporis of applications of models of collective phenomena in studying the "societies."
When assessing such models, I usually pay attention to the following issues:
1. The degree of simplifications of such societies.
2. Knowledge of the Authors of the relevant ideas from social sciences.
3. Checking whether, for studying such "society," the Authors do not simplify the behavior of units and collectivities.
4. We must remember that such models ALWAYS give results. Then savvy Authors can tell us their story describing and explaining the results. Often, they also can make predictions.
5. Considerations of cognitive aspects of behavior/interactions of actors.
6. It may be expected that in a not-so-distant future, all types of more or less abstract collectivities can be called social, and having enough more or less realistic data and computing power, the number of such texts will be infinite. This tendency is already visible in Entropy. The number of such papers may be expected to achieve millions.
7. On top of that, if the AI systems enhance their capabilities, that at present are only beginning, we may expect that future Authors only will define the topic and the boundary conditions concerning the "society." The rest will be written and…..published or not.
Bearing in mind the above, I have planned to prepare more on those aspects of this paper to facilitate the reviewing process. My present opinion is as follows:
1) My overall assessment is positive.
2) The text is well-designed, and the Author's competencies are very high.
3) However, from the above assumptions perspective, the paper has one significant weakness.
Although the Author is familiar with social theory, it concerns the collective models developed by social scientists.
The critical missing issue is perception. This term is used only twice in the text. To perceive - only once. The Author should share with the readers the reflections on how perception by the actors and their distortions can influence such models.
4) For example, the authors, fascinated by the mathematical form of the utility function, often forget about its origins. However, von Neumann declared it openly when he tried to develop a game theory to model actors' behavior based on subjective perceptions.
The paper can be published, but including the above issue would improve its academic quality.
Author Response
We wholeheartedly agree with the Referee's list of issues (1-7). In fact as far as issues 1-5 are concerned they should be taken into account when we prepare, conduct and eventually publish ABM-related studies. The call to create models with specific social issues in mind has been the basis for one of our earliest publications (Sobkowicz, P., 2009. Modelling opinion formation with physics tools: call for closer link with reality. Journal of Artificial Societies and Social Simulation, 12(1), p.11. Available at: http://jasss.soc.surrey.ac.uk/12/1/11.html.)
The issues 6 & 7 – forward looking – are equally important, though less realized. Creation of models for model’s sake made even easier by increases of computing power and the use of AI/ML may indeed create a deluge of such works (We wrote about it in Sobkowicz, P., 2020. Whither now, opinion modellers? Frontiers in Physics, 8, p.461.). It was very refreshing to find similar sentiments in the review of our paper. The improvements in data-driven, quantitatively accurate models and their predictions does not only impact their numbers but also allow for morally questionable (mis-)uses (Sobkowicz, P., 2019. Social Simulation Models at the Ethical Crossroads. Science and Engineering Ethics, 25, pp.pages143–157.) Similar (even more extreme) doubts exist when one applies AI/ML tools in social contexts (Sobkowicz, P., 2022. Hammering with the Telescope. Frontiers in Artificial Intelligence, 5. https://doi.org/10.3389/frai.2022.1010219).
To summarize, we fully appreciate and thank for the systematic approach taken by the Referee, sharing most of the issues used for the analysis.
In this context, the remarks of the Reviewer related to oversimplification and significant omissions (e.g. lack of consideration of the role of perception in opinion dynamics) are obviously true. However, the goal of the present contribution was quite humble: rather than to creata a detailed/complex model of individual and social behaviors, we used very simple models (separating the opinion and social network dynamics) to observe the effects of the positive role of local diversity of viewpoints/mindsets. We wanted to initiate studies of a system/situation with the goal of promoting socially beneficial activities – decreasing the polarization. The two models presented in the paper are but a first stage, showing that something along the proposed directions can be done.
This self-limitation notes were added in the introduction and conclusion sections (lines 56-60 and 594-600).
With respect to the use of the utility function, the motivation was to use a formulation that allows easier mixing of different approaches in a single model (e.g. bounded confidence or social influence formulations, external influences, etc.). The question of whether each agent uses the same utility function (and also to what extent the individual evaluations depend not on actual characteristics of the environment but on *perceptions* of these characteristics, filtered through individual biases) is left for the future studies.
Round 2
Reviewer 2 Report
I can only repeat my statements from my previous report:
This is an interesting paper that certainly can deserve publication. I however suggest various improvements. In my opinion the author took in consideration less than ½ of my suggestions. Referring to the items of my previous report:
-
It is still not stated what are the aim and novelty of the paper. The author should make these clear in the body of the papert and particularly in Abstract.
-
I still have a problem with mathematical formalism in Section 3.2. The symbols are not defined. What does it mean “n denotes neighbors”? What is n (I guess it should be an integer) and how it related to "neighbors"? What “n denotes neighbors” is written as a part of the equation? I am afraid that the author does not control the formalism simply because the formalism is too general.
-
The author agreed with me that the physical notions like “magnetic phenomena”, “temperature” in this context are just an effect of extrapolation but does not provide any remarks
5. The modelling in Section 5 is still rather mysterious. There are no details. The reader does not know what and how is simulated.
I am repeating my statement from the previous report
The paper needs an essential revision. The paper cannot have a visible impact in its present version. Only a paper in which ideas are clearly presented can have an impact. In this case the ideas are hidden.
Author Response
We thank the referee for her/his persistence: it forced us to re-read the paper with new eyes, and rework it.
With respect to point 1. We have rewritten the abstract, and included changes in Section 1, changing the “novelty” claims from the formal framework to simulation results and the motivation behind them. We agree that the framework (based on the utility function), while important in its capacity to combine very different mechanisms) is only a tool. The goal was an initial study of a subject long neglected (to our knowledge) in modeling literature, namely counter-polarization origins and mechanisms. We hope that with these changes the aim and novelty are sufficiently clear.
WRT 2: following the points raised by the Referee and the focus on results, rather than the intricacies of the framework we substantially rewrote section 3.2
WRT 3 We have included a dedicated discussion of the “mapping” of physical notions to social ones in the opening part of Section 4. In particular, it discusses the possibility of equating simulation (Monte Carlo) “temperature” with physical (thermodynamic) one, and the lack of a well-defined social counterpart.
WRT 5: we have expanded the opening part of Section 5 to include the description of the starting network configuration and the rationale for the choice (with key literature citations) and the description of the simulation process of rewiring of the network.
We hope that with these changes the paper would meet the referee standards.
Reviewer 3 Report
Given the authors’ description of their purposes in this paper, I think it is ready for publication if the reviewers with expertise in sociophysics regard it as sound. I felt that I could grasp the general logic better in this draft, although I still didn’t follow many of the details. I hope that someone (maybe the authors) will follow with a paper introducing this approach to readers who have background knowledge in public opinion but not in physics.
Reviewer 4 Report
I have no other comments. The explanation by the Author is sufficient.
Author Response
We thank the Referee for their comments.
Round 3
Reviewer 2 Report
Well, I understand the author did his best.